# MG53 suppresses interferon-β and inflammation via regulation of ryanodine receptor-mediated intracellular calcium signaling

Matthew Sermersheim [1], Adam D. Kenney[2], Pei-Hui Lin [1], Temet M. McMichael[2], Chuanxi Cai [1], Kristyn Gumpper [1], T. M. Ayodele Adesanya [1], Haichang Li[1], Xinyu Zhou [1], Ki-Ho Park [1], Jacob S. Yount[2] & Jianjie Ma [1]

TRIM family proteins play integral roles in the innate immune response to virus infection. MG53 (TRIM72) is essential for cell membrane repair and is believed to be a muscle-specific TRIM protein. Here we show human macrophages express MG53, and MG53 protein expression is reduced following virus infection. Knockdown of MG53 in macrophages leads to increases in type I interferon (IFN) upon infection. MG53 knockout mice infected with influenza virus show comparable influenza virus titres to wild type mice, but display increased morbidity accompanied by more accumulation of CD45+ cells and elevation of IFNβ in the lung. We find that MG53 knockdown results in activation of NFκB signalling, which is linked to an increase in intracellular calcium oscillation mediated by ryanodine receptor (RyR). MG53 inhibits IFNβ induction in an RyR-dependent manner. This study establishes MG53 as a new target for control of virus-induced morbidity and tissue injury.

[1] Department of Surgery, The Ohio State University Wexner Medical Center, Columbus, OH, USA. [2] Department of Microbial Infection and Immunity, The Ohio State University Wexner Medical Center, Columbus, OH, USA. ✉email: jacob.yount@osumc.edu; jianjie.ma@osumc.edu

Tissue repair in response to infection or injury is mediated by numerous different cell types. Macrophages are innate immune cells of myeloid origin that exhibit a wide range of functions and phenotypic heterogeneity. Macrophages are often classified as either canonically activated inflammatory macrophages (M1), or alternatively activated macrophages (M2) which are anti-inflammatory in nature and assist in tissue repair[1,2]. As such, macrophages play a pivotal role in the response to, and recovery from, infectious pathogens, such as viruses. Type I interferons (IFNs) are among the most important cytokines produced by inflammatory macrophages in response to viral infection and tissue injury. While a robust type I IFN response is critical for limiting the replication of viruses, when improperly regulated, IFN can also be maladaptive and pathologic[3]. As a result, cells possess coordinated mechanisms to fine tune the production of IFNs, and there is a growing appreciation of the importance of negative regulators of IFN induction[4].

Calcium is an essential regulator for nearly all physiological processes, including immunity and inflammation[5,6]. A means by which calcium elicits such wide effects is excitation−transcription coupling where rises in intracellular calcium oscillation drive gene transcription[5,7,8]. While several studies have reported the expression of ryanodine receptor (RyR) calcium channels in macrophages[9,10], the physiological role of RyR in immune regulation has yet to be fully characterized.

Tripartite motif-containing (TRIM) family proteins participate in the regulation of numerous physiological processes, including immunity[11]. The large family of over 70 proteins is characterized by an amino-terminal tripartite motif consisting of a RING domain, one or two B-box zinc finger domains, and a coil−coil region. The carboxyl-terminal region is variable among TRIM proteins allowing further classification into 11 subgroups, the largest of which is the class IV-PRY-SPRY domain-containing proteins. Several TRIM proteins are known to be regulated by viral infection and IFN[12,13], and many TRIM proteins participate in the control of viral immunity[14].

TRIM72, also known as mitsugumin 53 (MG53), is predominantly expressed in striated muscle[15]. Our laboratory has shown that MG53 is an essential component of cell membrane repair machinery[15–18]. Studies utilizing exogenous recombinant human MG53 (rhMG53) have shown protective effects against both mechanical injury and oxidative stress[15,18–21]. Surprisingly, the therapeutic potential of MG53 extends beyond the scope of cardiac and skeletal muscle, and also helps protect other tissues such as kidney, lung, brain, skin, aortic valves, and cornea[21–26]. Exogenous MG53 treatment has been shown to prevent fibrotic remodeling[21,27] and preserve lung function in mice during bacterial lipopolysaccharide (LPS)-mediated emphysema[23].

The wide therapeutic effects of MG53 in multiple tissues and in conditions in which pathology is mediated by inflammatory immune responses suggest that MG53 may have additional functions beyond membrane repair. While MG53 protein is mainly present in skeletal and cardiac muscles, our previous studies have shown that renal and alveolar epithelial cells also contain detectable amounts of MG53 that contribute to kidney and lung functions under physiological and pathological conditions[22,23]. Here we present data showing that human monocytic THP1 cells, as well as primary human blood monocyte-derived macrophages (hMDMs), express MG53 protein. Given the intimate association between innate immunity and tissue repair, we hypothesized that MG53 regulates inflammation during infection and tissue injury. We find that *MG53*-deficient mice display increased morbidity in response to influenza virus, which is characterized by exacerbated IFNβ production. We demonstrate that knockdown of MG53 results in aberrant macrophage calcium

handling, and that suppression of IFNβ transcription by MG53 is dependent on RyR-mediated calcium signaling.

## Results

**Human monocytes/macrophages express MG53**. MG53, once believed to be a muscle-specific TRIM protein, functions in membrane repair to help protect tissue from mechanical damage. Many TRIM proteins, especially those containing the PRY-SPRY motif at the carboxyl-terminus, are intimately involved in regulating inflammation and tissue repair[1,2,28]. Seeing that MG53 is a PRY-SPRY-containing TRIM protein involved in tissue repair, we sought to investigate whether MG53 is endogenously expressed in monocytes/macrophages.

THP1 cells are monocytic cells that can be differentiated into macrophage-like adherent cells, and are commonly used in the study of innate immunity. We first examined phorbol 12-myristate 13-acetate (PMA)-differentiated THP1 cells and observed MG53 expression visualized by western blotting (Fig. 1a). Immunoprecipitation with antibody against MG53 enhanced the intensity of the western blot band, whereas control IgG did not pull down MG53 protein. THP1 MG53 expression was notably lower than levels observed in murine heart and skeletal muscle (Fig. 1a). Western blot analysis shows that the MG53 protein level in THP1 cells is more than 400-fold lower than the level of MG53 in skeletal muscle (Supplementary Fig. 1). To test whether THP1 MG53 expression might be an artifact of immortalization, we also examined de-identified primary human macrophages. We similarly discovered endogenous MG53 expression in primary human blood monocyte-derived macrophages from distinct donors (Fig. 1b).

Type I IFNs are known to induce the expression of many TRIM proteins that contribute to antiviral defense[12,13] and viruses conversely antagonize the function of several TRIM proteins[14]. We thus examined whether MG53 expression is altered in THP1 cells upon infection with Sendai virus (SeV), a potent inducer of the innate antiviral immune response, including type I IFN[29] (Fig. 1c). We observed that SeV infection reduced MG53 protein expression by more than 50% (Fig. 1d). We also compared the effects of SeV and influenza virus H1N1 strain PR8 infection on MG53 expression in THP1 cells. As shown in Supplementary Fig. 2, while SeV infection consistently led to reduced MG53 protein levels in THP1 cells, influenza virus infection did not appear to induce a significant decrease in MG53 in THP1 cells. This indicates a potential virus-specific difference in the antagonism of MG53 expression in THP1 cells.

**Knockdown of MG53 results in an enhanced IFN and inflammatory response**. Appreciating that several other TRIM proteins are known to possess antiviral functions[30–34], we examined whether knockdown of MG53 affected infection rates of cells by SeV. We used shRNA to knock down the expression of MG53 in THP1 cells, and, in doing so, confirmed that MG53 is also expressed in undifferentiated THP1 cells (Fig. 2a). Control shRNA (sh-control) and sh-MG53 knockdown THP1 cells were infected with SeV expressing GFP for 24 h. Cells were collected and examined by flow cytometry for GFP fluorescence, indicative of virus infection and virus protein production (Fig. 2b). We observed that knockdown of MG53 did not significantly affect the percentage of cells infected with virus as compared to sh-control cells (Fig. 2c).

Having observed similar infection rates of sh-MG53 and sh-control THP1 cells, we next tested whether loss of MG53 altered the response of cells to virus infection. Specifically, we again utilized infection with SeV because of its potent ability to activate the innate immune response, including inflammatory cytokine

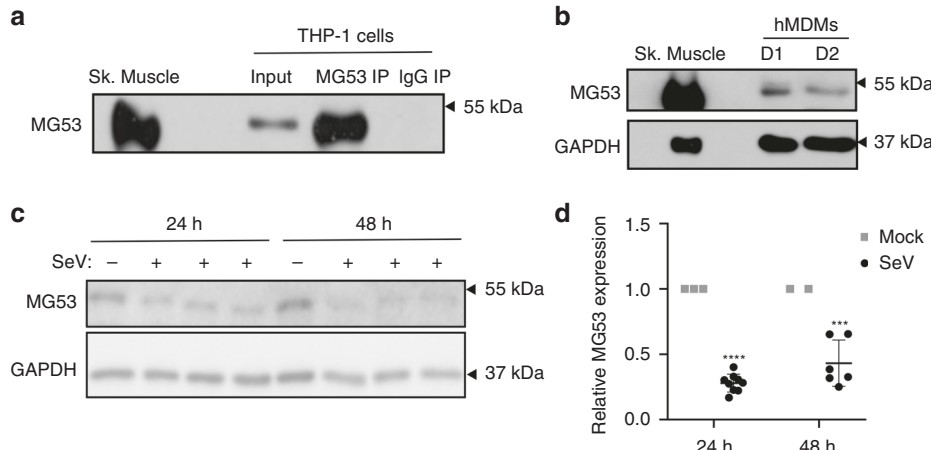

**Fig. 1 Viral infection reduces MG53 expression in macrophages. a** PMA-differentiated THP1 cells express endogenous MG53. MG53 protein in THP1 cells could be immunoprecipitated with anti-MG53 antibody. Mouse skeletal muscle (0.2 µg) was loaded as a positive control. THP1 protein lysate (20 µg) was loaded as input (data representative of three independent experiments). **b** De-identified primary human blood Monocyte-Derived Macrophages (hMDMs) also express MG53 protein. Murine skeletal muscle (0.5 µg) and 15 µg hMDM protein lysates (Donor 1/D1 and Donor 2/D2) were loaded for western blotting and MG53 immuno-detection (data representative of three independent experiments). Macrophage MG53 levels are lower than that seen in murine skeletal muscle (see also Supplementary Fig. S1). **c** Differentiated THP1 cells were infected with SeV (MOI 5) for 24 or 48 h. THP1 protein lysate (20 µg) was loaded for western blot and immune-probed for MG53 and GAPDH expression (data representative of three independent experiments conducted at 24 h and two independent experiments at 48 h). Infected cells had over 50% reduction in MG53 expression relative to noninfected controls. **d** Quantification of reduction in MG53 protein expression following SeV infection (data show three separate experiments conducted at 24 h and two separate experiments at 48 h, each experiment contained three independent infection replicates; mean ± SD; ***p = 0.0005, ****p < 0.0001; two-sided one-sample t tests).

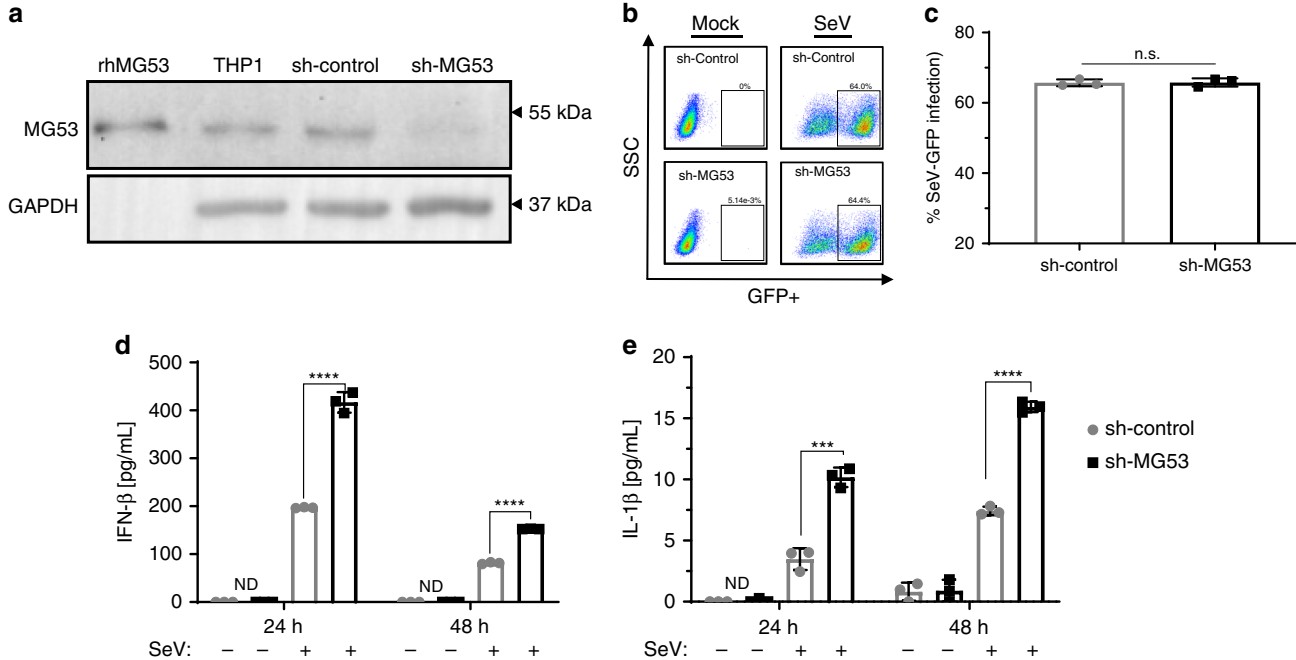

**Fig. 2 Knockdown of MG53 leads to a hyper-inflammatory response to viral infection. a** shRNA lentivirus was used to create stable sh-control and sh-MG53 knockdown THP1 cells. THP1 protein lysates (40 µg) were loaded for western blot and probed for MG53 and GAPDH expression. rhMG53 (0.1 ng) was loaded as a positive control (data representative of three independent experiments). **b** sh-control and sh-MG53 cells were infected with SeV-GFP (MOI 2) for 24 h. Cells were then analyzed for GFP+ signal via flow cytometry to determine the percentage of infected cells. There were no differences in infections rates between sh-control and shMG53 cells following SeV. **c** Quantification of percentage of SeV-GFP-positive cells (data representative of four independent experiments; mean ± SD; n.s. means nonsignificant, p = 0.9138; two-sided unpaired t test). **d, e** PMA-differentiated sh-control and sh-MG53 THP1 cells were infected with SeV (MOI 5) for 24 or 48 h. Supernatants were collected and assayed for cytokine secretion via ELISA. Knockdown of MG53 results in significant increase in IFNβ and IL-1β secretion 24 and 48 h after infection with SeV. Graphs depict representative data from three independent experiments, each performed in triplicate (mean ± SD; ***p = 0.0006, ****p < 0.0001, ND = not detected; two-sided unpaired t tests).

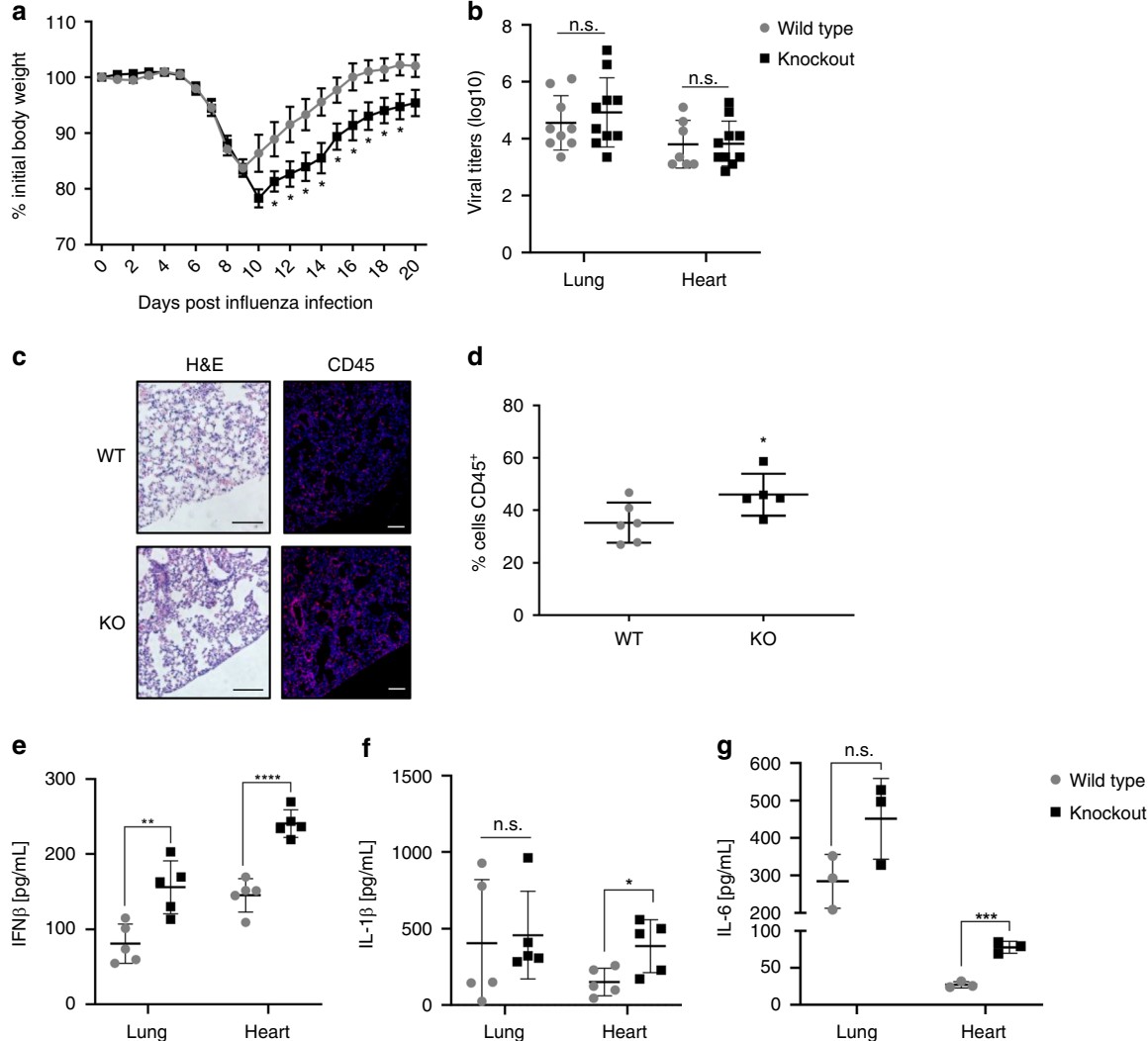

**Fig. 3 MG53 knockout mice have worsened morbidity following influenza infection. a** Mice were infected intranasally with influenza virus PR8 (10 TCID50). MG53 KO mice lost more weight and took longer to recover following PR8 infection compared to controls ($n = 12$ mice/group, mean ± SE; *$p < 0.05$ compared to WT animals at day post infection; multiple two-sided $t$ tests). **b** Lung and heart tissues were harvested from mice 5 days post infection. There was no significant difference in viral burden, as shown by comparable viral titers between WT and KO mice ($n = 9$ WT and 10 KO lungs; $n = 7$ WT and 10 KO hearts; mean ± SE; n.s. means nonsignificant; two-sided unpaired $t$ test). Lungs were isolated at 5 days post infection and either fixed for histology (**c, d**) or homogenized for viral titer and cytokine analysis (**e, f**). **c** H&E and CD45 staining of WT and KO lungs (images are representative of lung sections from six WT and five KO mice; scale bar = 100 μm). **d** Immunofluorescence staining revealed increased CD45+ leukocyte infiltration in MG53 KO lungs compared to WT ($n = 6$ WT and 5 KO mice; mean ± SD; $p = 0.0493$, two-sided unpaired $t$ test). **e** MG53 KO mice had increased IFNβ production in the lungs and heart ($n = 5$ mice/group; mean ± SD; **$p = 0.0051$; ****$p < 0.0001$; two-sided unpaired $t$ test). **f** Significant increases in IL-1β ($n = 5$ mice/group; mean ± SD; n.s. means nonsignificant $p = 0.8225$, *$p = 0.0281$; two-sided unpaired $t$ test) and **g** IL-6 ($n = 3$ mice/group; mean ± SD; ns nonsignificant $p = 0.0898$, ***$p = 0.0006$; two-sided unpaired $t$ tests) levels were measured in MG53 KO hearts compared to WT hearts.

production and secretion[29]. Despite similar infection rates (Fig. 2c), sh-MG53 cells yielded increased IFNβ and IL-1β secretion after SeV infection compared to sh-control cells (Fig. 2d, e). Thus, while MG53 does not alter SeV infection of cells, loss of MG53 results in a hyper-inflammatory cellular response to virus infection. These data indicate that MG53 may function to suppress type I IFN production and inflammation following viral infection.

**Increased morbidity of MG53 KO mice following influenza virus infection**. To examine whether MG53 plays a physiological role during in vivo viral infection, MG53 wild-type (WT) and knockout (KO) mice were intranasally infected with influenza virus strain PR8 at a dose of 10 tissue culture infectious dose 50 (TCID50). This dose causes weight loss in WT mice, peaking

around day 10, followed by a full recovery of body weight[35]. In MG53 KO mice, we observed a more severe decrease in weight following infection and a delayed recovery compared to WT mice (Fig. 3a).

To assess whether differences in virus replication and dissemination were responsible for worsened morbidity in KO mice, virus titers were measured from lung and heart tissues 5 days post infection. Interestingly, WT and KO mice showed no significant difference in virus titers across tissues (Fig. 3b), suggesting comparable levels of viral replication and dissemination at this time point post infection. However, KO animals had increased CD45+ immune cell infiltration within the lung environment (Fig. 3c, d). Similar to our in vitro experiments (Fig. 2c, d), we also observed a significant elevation in IFNβ levels in KO lungs (Fig. 3e). Consistent with the high levels of

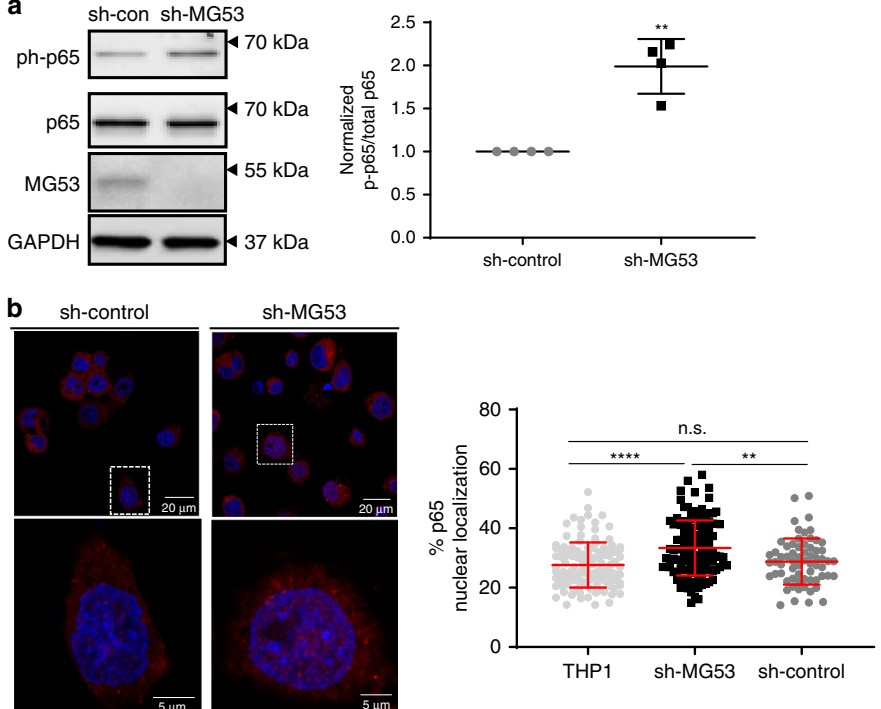

**Fig. 4 MG53 modulates NFκB activation and nuclear localization. a** Protein lysates (10 μg) from the indicated THP1 lines were loaded for western blot to assess p65 phosphorylation. Western blotting showed a ~2-fold increase in basal p65 phosphorylation in shMG53-THP1 cells compared with sh-control ($n$ of four independent experiments; mean ± SE; **$p = 0.0083$; two-sided paired $t$ test). **b** Cellular localization of p65 was assessed by immunofluorescence using p65 antibody. Nuclear localization of p65 is significantly increased in sh-MG53 THP1 cells compared to sh-control ($n = 119$ THP1, 119 sh-MG53, and 65 sh-control cells examined over two independent experiments; mean ± SD; n.s. means nonsignificant $p = 0.6300$, **$p = 0.0013$, ****$p < 0.0001$; one-way ANOVA with Tukey's multiple comparison test).

endogenous MG53 produced in muscle and heart tissue, IFNβ was also elevated in KO hearts along with an increase in the inflammatory cytokines IL-1β and IL-6 (Fig. 3f, g).

Cytokine levels presented in Fig. 3d–g are the actual concentrations within equal volumes of tissue homogenates. These measurements provide direct assessment of the physiological role of MG53 in modulation of inflammatory response following viral infection. The ~100% increase in IFNβ that we observe in infected MG53 KO lungs (Fig. 3e) is unlikely explained by the ~15% increase in infiltration of CD45$^+$ cells (Fig. 3d). Thus, the increased levels of IFNβ and inflammatory cytokines observed with the MG53 KO mice is unlikely a consequence of the higher number of immune cells present in those tissues.

Our data suggest that, despite similar viral burdens in WT and MG53 KO mice, the absence of MG53 leads to a maladaptive hyper-IFN response. These in vivo studies corroborate our in vitro findings and indicate that MG53 suppresses IFNβ production.

**MG53 modulates NFκB activation.** The mechanism by which MG53 mediates membrane repair has been thoroughly studied[15–17], but the means by which MG53 regulates inflammation is not known. The nuclear factor kappa-light-chain-enhancer of activated B cells (NFκB) family of proteins are critical transcription factors for induction of many inflammatory genes, and they are thus considered master regulators of inflammation. Among the best studied NFκB proteins is p65/RelA[36]. Upon infection, p65 is activated by a series of phosphorylation events that free it of inhibitory proteins and enable it to translocate to the nucleus to regulate gene transcription. We therefore examined whether knockdown of MG53 altered macrophage NFkB p65 activation. Under basal conditions p65 is normally unphosphorylated and localized to the cytoplasm in an inactive state.

We observed that, relative to sh-control THP1 cells, sh-MG53 cells displayed a near twofold increase in baseline phosphorylation of p65 at serine 536, a well-known NFκB activation marker (Fig. 4a). Immunofluorescent imaging revealed that along with elevated p65 phosphorylation, MG53 knockdown correspondingly resulted in more p65 localization to the nucleus (Fig. 4b). These data provide evidence that MG53 functions to negatively regulate NFκB activation and offer insight into how MG53 might regulate IFNβ and other inflammatory mediators.

**MG53 suppresses ryanodine receptor-mediated intracellular calcium release.** Previous studies have reported that MG53 can alter calcium handling in muscle cells by means of controlling store-operated calcium entry and suppressing RyR-mediated intracellular calcium release[37]. Rises in intracellular calcium oscillation and nuclear calcium can stimulate adaptor proteins to activate calcium-dependent transcription factors and drive transcriptional activation, a process known as excitation−transcription coupling[5,7,8]. We loaded sh-control and sh-MG53 THP1 cells with the calcium indicator Fluo-4-AM to test whether the loss of MG53 altered macrophage calcium dynamics. We found that MG53 knockdown yielded increased spontaneous calcium oscillations in macrophages (Supplementary Movie 1, Fig. 5a). To test whether the observed calcium oscillation was a result of intracellular calcium release from internal stores, we recorded cells in the presence of an extracellular calcium chelator, EGTA. In the presence of EGTA, we continued to observe spontaneous calcium oscillations in sh-MG53 THP1 cells (Fig. 5b). These data demonstrate that in the absence of MG53, macrophages exhibit dis-regulated intracellular calcium release.

There are predominantly two types of intracellular calcium release channels, inositol 1,4,5-triphosphate receptors (IP$_3$Rs)

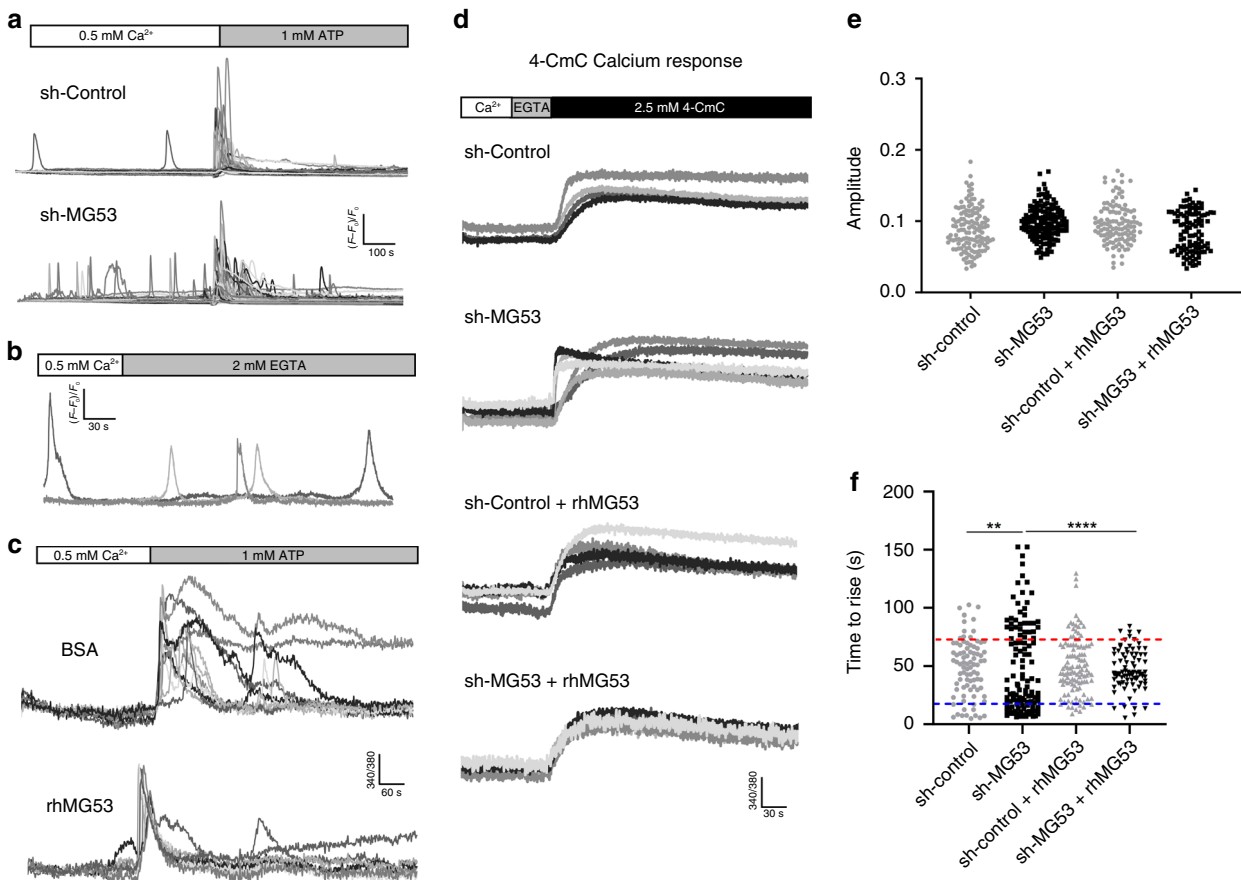

**Fig. 5 MG53 suppresses RyR intracellular calcium signaling. a** Representative traces of spontaneous calcium oscillations and ATP-invoked calcium release in sh-control and sh-MG53 THP1 cells. Approximately 10% of sh-MG53 THP1 cells exhibit spontaneous calcium mobilization compared to ~1% of sh-control cells (over 1400 cells, from multiple dishes, were imaged over three independent experimental days). See Supplementary Movies for live cell imaging of calcium oscillations. **b** Representative traces of spontaneous intracellular calcium release in shMG53-THP1 cells in the presence of 2 mM EGTA. **c** Calcium traces from differentiated THP1 cells loaded with Fura-2-AM. Cells were treated with either rhMG53 (1 μg/mL) or equal-molar BSA for 30 min and then imaged for calcium responses following 1 mM ATP stimulation. Exogenous rhMG53 treatment yielded more uniform calcium mobilization in response to ATP (data representative of three independent experiments). **d** 4-CmC-induced intracellular calcium release from sh-control and sh-MG53 THP1 cells, demonstrating bimodal activation kinetics in sh-MG53 cells. **e** Peak amplitude of 4-CmC-induced intracellular calcium release in individual sh-control and sh-MG53 THP1 cells was plotted with or without the addition of rhMG53 (1 μg/mL). **f** Time-to-rise (TTR) of 4-CmC-elicited intracellular calcium release in individual cells was plotted. Bimodal distribution of TTR was clearly seen in sh-MG53 THP1 cells, representing fast (denoted blue line) and slow components (denoted by red line) of 4-CmC-induced calcium release. Addition of rhMG53 reduced the population of sh-MG53 THP1 cells with fast component of 4-CmC-induced intracellular calcium release, with minimal impact on the sh-control cells ($n = 86$ sh-control, 122 sh-MG53, 114 sh-control + rhMG53, and 84 sh-MG53 + rhMG53 cells imaged over multiple dishes across five independent experimental days; cumulative distribution of data sets was analyzed and compared using Kolmogorov−Smirnov tests, **$p = 0.001$, ****$p < 0.0001$).

and RyR[38]. IP₃R channels are ubiquitously expressed across all cell types, while RyRs are canonically expressed in "excitable" cells, such as striated muscle and neurons[6]. Hosoi et al.[9] were the first to report that THP1 and numerous other immune cells express mRNA for RyR isoforms and are responsive to RyR agonist. Since then, others have confirmed these findings in multiple leukocytes including dendritic cells, and T- and B-lymphocytes[9,10,39–42].

We started our study of intracellular calcium release using ATP, a well-known activator of IP₃R-mediated calcium release[43]. We observed that sh-control THP1 cells exhibited transient increases in intracellular calcium levels, typical of ATP treatment (Fig. 5a). Interestingly, sh-MG53 cells often displayed prolonged, multiphasic oscillations in intracellular calcium following stimulation with ATP, indicating more active intracellular calcium release processes in the absence of MG53. Supportive of this notion, we observed that THP1 cells pretreated with rhMG53 exhibited more uniform calcium release following ATP treatment

when compared to bovine serum albumin (BSA)-treated controls (Fig. 5c).

Sh-control and sh-MG53 THP1 cells were next treated with the RyR agonist 4-chloro-m-cresol (4-CmC, 2.5 mM). Consistent with the report by Hosoi et al.[9] and Klegeris et al.[10], we observed release of intracellular calcium in sh-control THP1 cells following treatment with 4-CmC (Fig. 5d, first trace). In Supplementary Fig. 3, we showed that pre-incubation of THP1 cells in 20 μM of dantrolene (an inhibitor of RyR channel[44]) completely abolished 4-CmC-induced calcium release, confirming the role of RyR-mediated calcium release. MG53 knockdown THP1 cells did not exhibit considerable changes in the amplitude of 4-CmC-induced intracellular calcium release, but displayed a bimodal kinetic response to 4-CmC stimulation (Fig. 5e, f). The time in which it took cells to reach peak intracellular calcium levels (denoted time-to-rise) either occurred quickly (approximately 10 s) or gradually (taking approximately 50 s or longer) (Fig. 5d, second trace). Approximately 20% of sh-control THP1 cells responded to

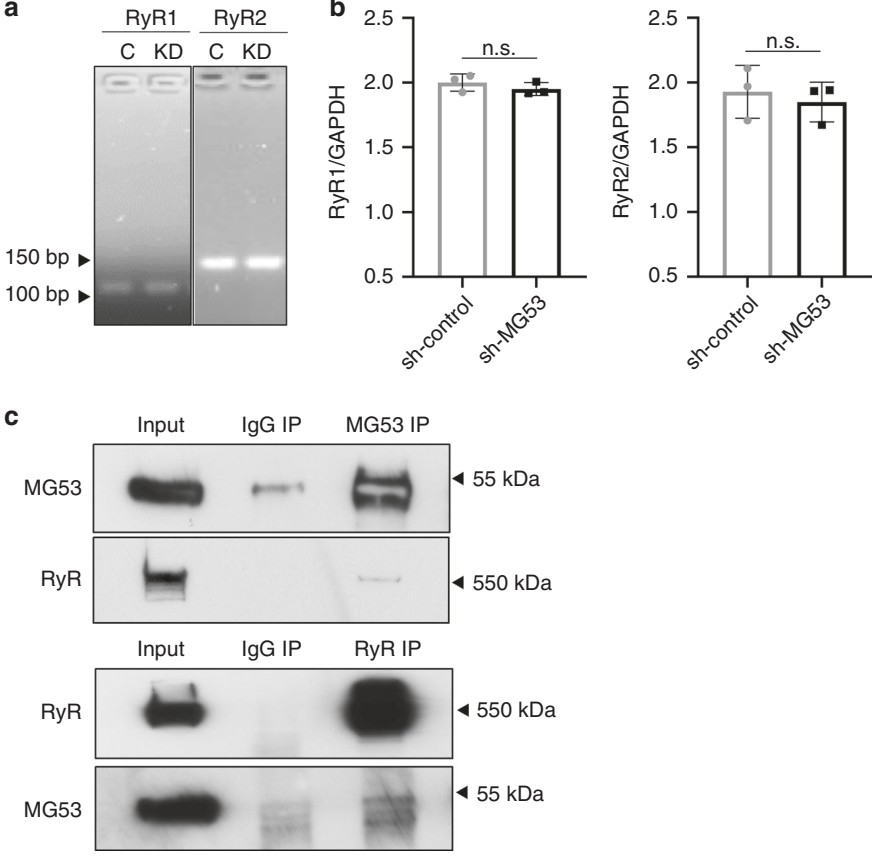

**Fig. 6 MG53 associates with RyR. a** Representative qPCR products of RyR1 and RyR2 from sh-control (lane labeled C) and sh-MG53 (lane labeled KD) THP1 cells (predicted amplicon size for RyR1 = 106 bp and RyR2 = 155 bp). **b** Quantification of RyR1 and RyR2 mRNA in sh-control and sh-MG53 cells shows no significant difference (data representative of three independent experiments each performed in triplicate; mean ± SD; n.s. means nonsignificant; two-sided unpaired *t* tests). **c** Murine skeletal muscle lysates were used for co-immunoprecipitation (IP) assay. IP with MG53 antibody pulled down RyR1 (top). Conversely, IP with RyR antibody pulled down MG53 (bottom). Data are representative of four independent experiments.

4-CmC with quick calcium release. However, following knockdown of MG53, 49% of the MG53 knockdown cells displayed a propensity for quick calcium release (Fig. 5d, f). Treatment of sh-MG53 cells with rhMG53 abrogated the bimodal distribution of 4-CmC-induced intracellular calcium release, providing a result similar to the control THP1 cells (Fig. 5d, third and fourth trace). As a result of rhMG53 treatment, sh-MG53 THP1 cells exhibited more uniformity in calcium release following 4-CmC activation. Overall these data indicate that loss of MG53 increases RyR activity in macrophages, a phenotype which is reversible with exogenous rhMG53 treatment.

**MG53 associates with ryanodine receptor complex**. Alterations in RyR calcium dynamics could possibly be attributed to increases in RyR expression with MG53 knockdown or changes in the regulation of RyR gating. To address these possibilities, we compared RyR1 and RyR2 mRNA levels between sh-control and sh-MG53 THP1 cells using quantitative real-time PCR. Knockdown of MG53 did not lead to significant changes in THP1 RyR mRNA expression (Fig. 6a, b).

We next tested whether MG53 might directly or indirectly complex with RyR to potentially regulate its function. THP1 cells express very low levels of RyR, which made immunoprecipitation of the protein technically unfeasible in this cell type. Therefore, we performed co-immunoprecipitation assays using murine skeletal muscle, a tissue in which both proteins are abundantly expressed. As shown in Fig. 6c, immunoprecipitation with antibody against MG53 also precipitated RyR, and immunoprecipitation with

antibody against RyR precipitated MG53, indicating that MG53 and RyR are present in a molecular complex with one another. While our co-IP studies demonstrate the co-presence of MG53 and RyR1 in a complex within skeletal muscle, whether or not such MG53-RyR1 interaction is direct or requires the participation of an intermediate protein component requires further studies.

**Suppression of IFNβ transcription by MG53 is RyR-dependent**. Intrigued by the reduction of macrophage MG53 expression yielding both aberrant calcium signaling and increased IFNβ secretion following viral infection, we examined whether MG53 regulates IFNβ transcription and if this is dependent on RyR-calcium signaling.

Our initial studies, using HEK293 cells transfected with IFNβ-luciferase reporter plasmid along with either GFP control or MG53, revealed that MG53 did not affect IFNβ-luciferase reporter activity (Fig. 7a). This phenomenon was observed under both basal and SeV-infected conditions. However, HEK293 cells are known to lack expression of RyR channels[45,46].

To determine whether MG53 regulation of IFNβ was RyR-dependent, we implemented the use of IFNβ-luciferase reporter in HEK293 cells engineered to express RyR$_2$ under the control of a doxycycline-inducible promoter[47]. In the presence of RyR, coexpression of MG53 decreased IFNβ-luciferase production with or without SeV infection (Fig.7b). To further elucidate the mechanism by which MG53 alters IFNβ transcription, we tested the effect of RyR and MG53 on known transcription

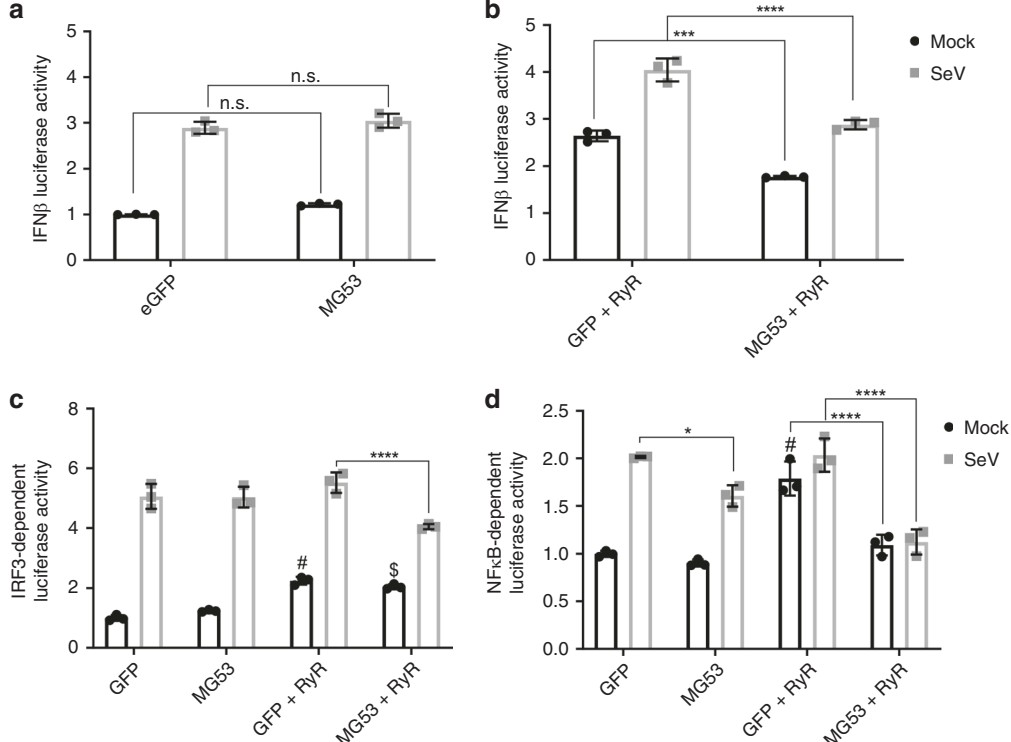

**Fig. 7 MG53-mediated suppression of IFNβ transcription is RyR-dependent. a** HEK293 cells were transfected with IFNβ luciferase and control Renilla luciferase reporters in addition to GFP or MG53 expression constructs. Cells were then either mock treated or infected with SeV (MOI 5) for 8 h. Overexpression of MG53 did not significantly affect IFNβ Luciferase activity with (gray) or without SeV infection (black) (data representative of three independent experiments; mean ± SD; n.s. means nonsignificant; two-way ANOVA with Sidak's multiple comparison test). **b** HEK293 cells expressing RyR2 were used to assess IFNβ luciferase activity. Cells were transfected with luciferase reporters in addition to either GFP or MG53 expression constructs. Cells were mock treated or infected with SeV. MG53 significantly suppresses luciferase activity in HEK293 cells expressing RyR (data representative of three independent experiments; mean ± SD; ***$p = 0.0002$, ****$p < 0.0001$; two-way ANOVA with Sidak's multiple comparison test). **c** Overexpression of MG53 suppresses SeV-induced IRF3 transcriptional activity when RyR is coexpressed (data have been normalized to mock GFP group and is representative of three independent experiments; mean ± SD; ****$p < 0.0001$, "$" denotes $p = 0.0016$ relative to mock GFP, "#" denotes $p = 0.0002$ relative to mock GFP; two-way ANOVA with Sidak's multiple comparison test). **d** Under mock and infected conditions, overexpression of MG53 suppresses the agonistic effect of RyR on NFκB reporter activity (data have been normalized to mock GFP group and is representative of three independent experiments; mean ± SD; *$p = 0.0148$, ****$p < 0.0001$, "#" denotes $p < 0.0001$ relative to mock GFP; two-way ANOVA with Sidak's multiple comparison test).

factor-specific domains within the promotor of IFNβ. IFNβ transcription is dependent on IFN Response Factors 3 and 7 (IRF3/7) as well as NFκB binding. IRF3/7 recognize and bind to two regions of the IFNβ promotor known as PRD-I and PRD-III, while NFκB enhances IFNβ transcription via binding along the PRD-II domain. With the use of 3xPRD-I and 4xPRD-II specific luciferase reporters[48], we tested the effects of RyR and MG53 on IRF3/7- and NFκB-dependent transcriptional activity. Overexpression of MG53, in the presence of RyR, suppressed the activity of both IRF3 and NFκB following SeV infection (Fig. 7c, d), confirming the ability of MG53 to inhibit the IFN response to infection.

## Discussion

Proper balance and regulation of innate immunity is imperative to human health, including during virus infection. Type I IFNs play a crucial role in virus restriction and host antiviral defense. However, when left unchecked, IFN production can be pathological by promoting excessive inflammation, known as cytokine storm, which leads to tissue damage[3,49,50]. The duality of IFN during viral infection necessitates numerous negative regulatory mechanisms within IFN pathways[4]. With the growing notoriety of TRIM proteins and their roles in immunity, our study now demonstrates a new immune-regulatory role for MG53. We discovered that MG53 prevents a maladaptive hyper-inflammatory

response characterized by excessive IFNβ production during viral infection. We further characterize the cellular mechanism underlying MG53-mediated suppression of IFNβ transcription and inflammatory signaling in association with intracellular calcium release.

Macrophages play an important role in the detection of and defense against pathogens such as viruses. We identified that human macrophages endogenously express MG53, and that MG53 expression is reduced following certain viral infections, such as SeV. This finding was surprising given that many TRIM family proteins are upregulated by IFN signaling[12,13]. A study performed by Tran-Thi et al.[51] in porcine skeletal muscle satellite cells supports our data. They identified MG53 as a major gene downregulated by TLR3 agonist, and viral mimic, poly(I:C). Their interpretation was that viral infection downregulates genes involved in satellite cell differentiation. However, when taken into consideration with our findings in macrophages, it is possible that cells downregulate MG53 expression during viral infection to allow a robust IFN response. Our findings indicate that, beyond membrane repair, MG53, like other TRIM proteins, has a distinct immunological function.

Both in vitro and in vivo studies demonstrate that loss of MG53 did not significantly alter rates of viral infection, but instead promotes a hyper-inflammatory response to viral infection. Specifically, we observed that animals and cells lacking

MG53 exhibit significantly elevated levels of the IFNβ, IL-1β, and IL-6 in response to influenza virus or SeV infection. Enhanced inflammation associated with the absence of MG53 is attributed to a significant increase in NFκB activation, i.e., increases in p65 phosphorylation and nuclear localization. In vitro studies using macrophages infected with virus mimic an inflammatory M1 activation of macrophages. Our findings show that suppression of MG53 exacerbates inflammatory cytokine secretion following viral infections, suggesting that MG53 functions to negatively regulate M1 macrophages. In vivo studies in which mice were infected with influenza virus supports the notion of MG53 suppressing M1 macrophages because they too yield elevated inflammatory cytokine production in the absence of MG53. However, due to the exaggerated recovery time of influenza-virus-infected MG53 knockout mice, it is possible that MG53 might also act to regulate M2 macrophage function and tissue repair. Further studies are required to fully detail the impact of MG53 expression on different macrophage phenotypes and their unique functions.

Calcium is a well-known regulator of many cellular processes, including gene transcription. The importance of calcium regulation in control of inflammation and immunity is becoming increasingly apparent. The role of calcium signaling in T-cell-receptor- and B-cell-receptor-mediated lymphocyte activation has been well documented over the years[41,42,52,53]. Murakami et al.[54] showed that both extracellular calcium entry and intracellular calcium release are important during inflammasome activation and subsequent IL-1β processing. It has previously been reported that STING-mediated induction of IFNβ is blocked by intracellular calcium chelation[55]. Similarly, others have reported that LPS-mediated IRF3 activation is dependent upon phospholipase Cγ2 and IP3R intracellular calcium signaling[56].

While it is known that MG53 regulates skeletal muscle RyR calcium release[37], we are the first to show that MG53 suppresses macrophage RyR-mediated intracellular calcium handling. RyR is predominantly expressed in excitable tissue. However, similar to what others have published[9,10], we report mRNA expression of RyR1 and RyR2 in THP1 cells. Immune cell RyR expression is much lower than that seen in skeletal and cardiac muscle. The Human Protein Atlas consensus dataset reports that, relative to skeletal muscle, immune cells such as monocytes and dendritic cells express 2.7% and 2.3% of the amount of RyR1 RNA, respectively[57]. Since knocking down the expression of MG53 impacted intracellular calcium oscillation in THP1 cells, in principle other calcium-dependent genes may also be altered which can contribute to the increased inflammatory response of the THP1 cells following viral infection. Additional genomic, biochemical, and functional studies will be required to address this topic. Earlier studies by Treves and colleagues demonstrated that mutations of RyR1 in human patients with malignant hyperthermia and central core disease impacted IL-6 release from skeletal muscle[58], which is consistent with the role of RyR-mediated calcium release in modulation of inflammatory cytokine release.

We demonstrate here that MG53-mediated suppression of IFNβ transcription occurs in an RyR-dependent fashion. MG53 suppresses IFNβ production via RyR-dependent inhibition of IRF3- and NFκB-dependent gene expression. Recently, two separate groups have reported that MG53 suppresses NFκB activity in neonatal rat ventricular myocytes and hippocampal neuronal cells, respectively[59,60]. It is of note that both experimental models express high levels of RyR isoforms. Liu et al.[59] reported MG53 co-immunoprecipitates with TGF-β activated kinase 1 (TAK1) and nuclear factor of kappa light polypeptide gene enhancer in B-cells' inhibitor, alpha (IκBα), two important proteins involved in NFκB signaling. This provides evidence that MG53 and RyR may function at multiple levels within the NFκB signaling cascade.

Questions remain regarding the global implications that MG53-mediated calcium alteration might illicit on other cellular processes, such as proliferation and cell survival/death. Additionally, the known importance of calcium to both T-cell receptor and B-cell receptor signaling would make an interesting study on the expression of MG53 in other leukocyte populations, particularly lymphocytes, and its potential impact on regulating leukocyte-specific cellular functions. It is important to note that lymphocytes also express RyR. RyR-mediated calcium signaling appears to be an important means of sustaining TCR activation and IL-2 signaling[9,39,61,62]. The data presented suggest MG53 could additionally play an exciting role in regulating RyR-mediated lymphocyte activation/maturation and is worthy of further investigation[9,39,61,62].

The original discovery of MG53-mediated membrane repair was a landmark finding in cell and tissue repair[15]. Our new work suggests that membrane repair is the "tip of the iceberg" in the study of MG53. The present study identifies a new function for MG53 in immunity. This work will hopefully inspire further research on how MG53 and other TRIM proteins function to regulate immunity, infection, and disease.

## Methods

**Animal handling and mouse influenza infection**. MG53 knockout mice were generated in the 129S1/SvlmJ strain of mice[15]. Mice have been backcrossed and maintained for over 30 generations and used in numerous tissue injury studies[19,21–23,25,26,63]. All mice were housed and handled in the Ohio State University Laboratory Animal Resources vivarium, with temperature, humidity, and light/dark schedule controlled following Ohio State University Institutional Animal Care and Use Committee-approved protocols in accord with National Institute of Health guidelines. Murine intranasal influenza virus infections were carried out in 12-week-old male MG53 wild-type and knockout mice. Animals were anesthetized using isoflurane and were intranasally infected with influenza virus strain A/PR/8/34 (H1N1) (PR8) at a dose of 10 tissue culture infectious dose 50 (TCID50) in 50 μL clinical grade saline. Mice were monitored daily and weights were recorded. Animals were euthanized at either day 5 post infection or at the experimental endpoint when they recovered to normal body weight. After animals were sacrificed, lungs and hearts were collected for viral titers, cytokine measurements, and histology.

**Western blotting**. Cells and murine skeletal muscle were lysed in radio-immunoprecipitation assay lysis buffer (Alfa Aesar, J63306) containing protease and phosphatase inhibitors. Cellular debris was pelleted via centrifugation and supernatants were collected for protein quantification by Bradford assay. Samples were prepared in 2× Laemlli sample buffer and separated on SDS-PAGE gels via electrophoresis, followed by wet transfer onto polyvinylidene difluoride membrane. Membranes were blocked in 5% milk in TBS-T and probed with antibodies against MG53 (custom-made rabbit monoclonal antibody, 1:1000)[22,25], glyceraldehyde 3-phosphate dehydrogenase (GAPDH; Cell Signaling Technology (CST) catalog # 2118) 1:3000, p65 (CST 8242) 1:1000, phospho-p65 (CST 3033) 1:1000, or RyR (Invitrogen MA3925) 1:1000.

**Co-immunoprecipitation**. Tissue and cells were lysed in radio-immunoprecipitation (IP) assay lysis buffer and assayed for protein concentration as stated above. Magnetic protein G beads (per IP sample) (20 μL) were washed in phosphate-buffered saline (PBS) three times and conjugated to 2 μg (per IP sample) of antibody (MG53, RyR, Mouse and Rabbit IgG) for 2 h at room temperature while rocking. Bead−antibody conjugates were then washed two times with PBS and once with lysis buffer. One milligram of protein lysate was added to beads and then samples were incubated at 4 °C overnight while rocking. The following day, samples were washed three times in PBS and protein was eluted with 4% SDS and 2× Laemlli sample buffer. IP samples were then analyzed following the western blotting protocol stated earlier.

**Cell culture**. THP1 cells were purchased from ATCC and cultured in RPMI-1640 media supplemented with L-Glutamine and sodium pyruvate (Sigma R8758) in addition to 10% fetal bovine serum and 1% penicillin/streptomycin in a 5% $CO_2$ incubator. THP1 cells were differentiated using 100 ng/mL PMA (Sigma P1585) for 48 h. HEK293 and HEK293FT cells were obtained from ATCC and Thermo Fisher, respectively, and cultured using Dulbecco's modified Eagle's medium (DMEM) supplemented with 10% fetal bovine serum and 1% penicillin/streptomycin in a 5% $CO_2$ incubator. HEK293-RyR2 cells were provided by Dr. Wayne Chen[47]. These

cells possess doxycycline-inducible RyR2 expression, which enables spontaneous calcium oscillation in response to elevated extracellular calcium via store-overload induced calcium release[47,64–66]. Cells were cultured using DMEM supplemented with 10% fetal bovine serum and 1% penicillin/streptomycin in a 5% $CO_2$ incubator. Treatment with doxycycline (1 μg/mL) for 24 h was used to induce RyR expression.

**Knockdown of MG53 in THP1 cells**. Control shRNA (5′-GACTGACATGTCAA GCTGTAC-3′) and MG53 shRNA (5′-GAAGAGTGTGGCTGTGCTGGAGCATC AGC-3′) were ligated into pKLO-mcherry-puro vector. In brief, HEK293-FT cells were transfected with packaging, envelope, and target plasmids. Media was changed 18 h after transfection, followed by collection of virus-containing media 48 h later. Virus-containing media was centrifuged at $1200 \times g$ for 5 min and filtered with 0.45-μm filters. THP1 cells were then incubated with viral media. After 24 h, media was replaced, and cells were allowed 48 h to recover. Following recovery, cells were selected for using puromycin (1.0 μg/mL), and subsequently cultured in RPMI-1640 media supplemented with puromycin (0.5 μg/mL), to generate sh-control and sh-MG53 THP1 cells.

**Viruses, in vitro Infections, and flow cytometry**. SeV expressing GFP, SeV strain Cantell, and influenza virus strain PR8 were propagated in embryonated chicken eggs and titered on LLCMK2 cells for SeV and MDCK cells for influenza virus as described previously[67]. SeV-GFP and SeV infections were allowed to proceed for 24 or 48 h using multiplicity of infections (MOIs) of 2 and 5 respectively. Twenty-four hours post SeV-GFP infection, THP1 cells were washed in PBS and fixed using 4% paraformaldehyde. Cells were washed, resuspended in PBS, and analyzed with a FACSCanto II flow cytometer (BD Biosciences) to determine the percentage of GFP-positive cells. Gating was performed to remove cellular debris and to ensure analysis was performed only on singlet cells as determined by forward and side scatter measurements (Supplementary Fig. 4). Data were analyzed using FlowJo software.

**Cytokine quantification**. THP1 cells were infected with SeV for 24 and 48 h. Following infection, supernatants were collected and centrifuged at 1000 rpm to remove cellular contamination. Cleared supernatants were flash frozen in liquid nitrogen and stored at −80 °C. For murine tissue samples, organs were necropsied, and homogenized in PBS containing protease inhibitor cocktail. Homogenates were cleared via centrifugation, flash frozen in liquid nitrogen, and stored at −80 °C. Cell supernatants and tissue homogenates were assayed for cytokine levels via enzyme-linked immunosorbent assay (ELISA), using human or mouse IFN-β, IL-1β, and IL-6 ELISA kits (R&D Systems, DY814, DY201, DY8234, DY401, and DY406). Experiments were performed following the manufacturer instructions.

**Quantitative PCR to characterize RyR expression in THP1 cells**. Total RNA was isolated from cells using Qiagen miRNeasy kit (217004). RNA quantity and quality were assayed using a Nanodrop 2000 spectrophotometer (Thermo-Fisher). RNA was then converted to cDNA using an iScript cDNA synthesis kit (Bio-Rad, 1708891). Once cDNA was made, qPCR was run on a CFX Real-Time PCR thermocycler (Bio-Rad) using human RyR1 (Bio-Rad, 10025636), RyR2 (F: GAG GGACTTCCACAAAGCGA and R: GGCATGTGCTCAGAGAGGTT), and GAPDH (F: GAAGGTGAAGGTCGGAGTC and R: GAAGATGGTGATGGGAT TTC) primers. Quantification was done using CFX Maestro software (Bio-Rad).

**Immunofluorescence staining**. THP1 cells were cultured on 35-mm petri dishes with a no. 1.5-cm glass coverslip window. Cells were fixed in 4% paraformaldehyde for 20 min, followed by simultaneous permeabilization and blocking with blocking buffer consisting of 0.2% triton X-100 in 3% BSA (w/v) in PBS for 1 h at room temperature. Primary antibody to p65 was diluted 1:200 in blocking buffer and incubated overnight at 4 °C. Cells were later washed three times in PBS (allowing 5 min per wash) and labeled with secondary antibody conjugated to Alexa-647 for 1 h at room temperature. Nuclei were stained with 4′, 6′ diamidino-2-phenylindole dihydrochloride(DAPI) (1:5000) followed by washing three times. Antibodies were "locked" into place with a 5-min post-fixation in 4% paraformaldehyde. Images were taken using a Zeiss 780 confocal microscope.

**Histology and immunohistochemistry**. Mouse lungs were harvested and fixed in 4% paraformaldehyde overnight, followed by dehydration in 70% ethanol for 24 h. Lungs were then transferred to The Ohio State College of Veterinary Medicine Histology Laboratory for paraffin embedding, sectioning onto slides, and staining with hematoxylin and eosin. Slide sections were imaged on an Axiovert 200 microscope (Zeiss) with CellSens Dimension 1.18 imaging software (OLYMPUS). Four-micrometer-thick paraffin sections were cut onto slides for immunofluorescent staining, which was performed as follows: slides were deparaffinized and rehydrated by incubating successively in xylene, 100% ethanol, 95%, 75%, 50% ethanol, and PBS. Antigen retrieval was achieved by heating in a pressure cooker with Tris-EDTA buffer for 13 min. Primary antibody (CD45, Abcam ab10558, 1:200) was applied and incubated at 4 °C overnight. DAPI was used to counterstain the nuclei of cells. All images were captured on a Nikon A1R confocal microscope and analyzed by ImageJ.

**Luciferase reporter assays**. Cells were transfected with pCMS-eGFP, pCMS-myc-MG53/GFP, IFNβ-Firefly Luciferase, PRDI-Firefly Luciferase, PRDII-Firefly Luciferase[48], and pRL-TK (Promega) expressing Renilla Luciferase using PolyPlus JetPrime transfection reagent. Luciferase reporters were kindly provided by Dr. Dimitris Thanos (Biomedical Sciences Research Center, Athens, Greece)[68]. After allowing 24 h for transient expression, cells were infected with SeV at an MOI of 5. Eight hours after infection, cells were lysed and assayed for luminescence using Promega Dual-Luciferase Reporter kit (E1910) and a Veritas microplate luminometer. All experiments were performed in triplicate, with Firefly luciferase signal normalized to Renilla luciferase.

**Calcium imaging**. THP1 cells were cultured on 35-mm petri dishes with a no. 1.5-cm glass coverslip window. Cells were washed in a balanced salt solution containing: 140 mM NaCl, 2.8 mM KCl, 2 mM $MgCl_2$, 0.5 mM $CaCl_2$,12 mM Glucose, and 10 mM HEPES (pH 7.4, mOsm = 290 ± 10). Cells were treated with either 2.5 μM Flou-4-AM or Fura-2-AM for 45 min at 37 °C in a 5% $CO_2$ incubator. Experiments were carried out in the presence or absence of 2 mM EGTA. Fluo-4 imaging was done using a Zeiss 780 confocal microscope, while the ratio of Fura-2 fluorescence at excitation wavelengths 340 and 380 nm was measured on a PTI spectrofluorometer (Photon Technology International)[69,70].

**Statistical analysis**. All statistical analyses were performed using GraphPad Prism Software (version 8.1.2). Specifics regarding results and the statistical tests performed are provided in each figure legend.

**Reporting summary**. Further information on research design is available in the Nature Research Reporting Summary linked to this article.

## Data availability

The authors declare that the data of this study are available within the article and its supplementary information files. Human RyR RNA expression comparisons were done using The Human Protein Atlas at https://www.proteinatlas.org/ENSG00000196218-RYR1/tissue. Source data are provided with this paper.

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

## Acknowledgements

We thank Dr. Wayne Chen for providing the doxycycline-inducible RyR2-expressing HEK293 cells, Drs. Juan Moliva and Jordi Torrelles for providing the primary human

blood monocyte-derived macrophage protein lysates, and Dr. Dominique Garcin for providing SeV-GFP. This work was supported by NIH grants AR061385, AR070752, DK106394, and HL138570 to J.M. and AI130110 and AI142256 to J.S.Y. M.S. was the recipient of a Presidential Fellowship from The Ohio State University. A.D.K. was supported by an NIH T32 fellowship funded by grant GM068412.

## Author contributions

J.M. and J.S.Y. designed the research. M.S., A.D.K., P.-H.L., T.M.M., C.C., K.G., T.A.A., H.L., X.Z. and K.-H.P performed research and data analysis. M.S., J.S.Y,and J.M. wrote the manuscript.

## Competing interests

J.M. has an equity interest in TRIM-edicine, which develops MG53 for the treatment of human disease. Patents on the use of MG53 are held by Rutgers University Robert Wood Johnson Medical School. All other authors confirm no other conflicts of interest.
