## [Peer Review File · Nature Communications]

Reviewers' comments:

Reviewer #1 (Remarks to the Author):

The authors are the leaders on the TRIM72 area of research. TRIM72 is also known as MG53. The new findings are truly exciting.

TRIM72 is essential for cell membrane repair and is believed to be a muscle specific. The group now has convincing data to demonstrate that human macrophages express MG53, and that MG53 protein is reduced following viral infection. Knockdown of MG53 in macrophages leads to increased type I interferon (IFN) production upon infection, implicating MG53 as a negative regulator of IFN response. They further demonstrate a specific interaction into of MG53 signaling in activation of NF κ B signaling, which is linked to an increase in intracellular calcium oscillation mediated by ryanodine receptor (RyR). Last, they show that MG53 interacts with RyR and inhibits IFN β transcription in an RyR-dependent manner.

This study establishes MG53 as a new macrophage target for control of virus-induced disease and tissue injury. It can have broad implications for public healthcare.

I only have very minor comments:

a) Add a couple of sentences about the overall role of macrophages on inflammation and the switch in M1 to M2 for resolution of inflammation.

b) Item a above, do the same in the discussion. Could speculate whether MG53 interaction occurs with M1 or M2, likely M2.

c) The data is very robust and it is excellent to see all data points plotted for transparency. We do recommend that for some of the data, the authors further check the statistics with non parametric tests and ad hoc adjustments.

d) The authors could test using either Pearson or Spearman whether there is a direct correlation between MG53 levels and immune activation or disease state.

Reviewer #2 (Remarks to the Author):

In their paper the Authors describe a novel role for the skeletal muscle protein MG53 as a mediator of innate immune response. The Authors show that MG53 is expressed in human macrophages where it regulates the expression of interferon. Knocking down MG53 leads to increased interferon production via NF κ B signaling. The increased nuclear localization of NF κ B is mediated by RyR-dependent calcium oscillations.

These observations are novel and interesting, especially because until now the primary role of MG53 was thought to be primarily tissue (skeletal muscle) repair.

In my opinion additional experiments need to be performed in order to be convincing.

Major comments

1) In Figure 1 the Authors show by IP that macrophages express MG53. Can they see an immunopositive band by simple Western blot and not only after immunoprecipitation? No information regarding the total amount of protein loaded is indicated. How many cells were lysed? How many micrograms of proteins were loaded on the gels? They state that the levels are much lower than in skeletal muscle, but this is not quantified. Quantification by RT-qPCR or western blot comparing the intensity of the immunopositive band in skeletal muscle, cardiac muscle, macrophages and several other tissues is necessary. No explanation for panel 1D is included. How

were the values normalized?

2) In figure 2 and figure 3 they show that knocking down/out MG53 leads to an increase in IFN production, yet mice infected with influenza virus do not show a difference in viral load. How come? IFN is released by cells to counterbalance viral infections and thus higher levels of IFN should be associated with a decreased viral load and lower morbidity. Mice should recover faster and not slower if they have an enhanced inflammatory response. Production of additional cytokines should be assessed including IL6, TNF α , IL12, IL23 as well as chemokines. Phenotypically distinct classes of macrophages exist, M1 which are pro-inflammatory and microbicidal and M2 which are immunosuppressive. Does the absence of MG53 affect all macrophage types equally? Please also add serum levels of CRP as this is a good indication of the inflammatory response.

3) The fact that lungs and heart show increased levels of IFN and IL1 may simply be a consequence of the higher number of macrophages present in those tissues. Can the levels of these cytokines be normalized to number of macrophages present? As shown in figure 3 levels of the cytokines are not normalized to anything.

4) The involvement of RyR in calcium homeostasis in THP cells is not convincing. Extremely high levels (2.5 mM) of 4-CmC were used. The Authors need to show that there is an effect on calcium release following the addition of 300-600 μ M 4-CmC or 5-10 mM caffeine. They should also show that this effect can be blocked with dantrolene, procaine or by pre-incubating the cells with high levels of ryanodine as well as in cells silenced for RyR1 and RyR2.

5) The qPCR results show that RYR1 and RYR2 are expressed in THP cells and that their levels of expression are not affected by MG53 levels. What are the levels of RYR1 and RYR2 expression in THP cells compared to heart, skeletal muscle, brain, cerebellum, smooth muscle cells and other immune cells?

6) It is puzzling that MG53 and RyR1 form a complex since to my understanding MG53 is in a different compartment (either cytosolic or sarcolemmal) while the RyR1 is in the sarcoplasmic reticulum. Does MG53 form a complex with the DHPR? and junctophilin? Because MG53 is a TRIM protein, it may be sticky and bind non-specifically to many proteins in vitro but this may never happen in vivo.

Minor

1) Statistical analysis tests are not always defined (see for example figure 4, 5, 7, which post hoc test was used after the ANOVA? What was used in figure 5F?).

2) Many panels in the figures are not described. Please add the missing information to panels 1D, 2C.

3) For all experiments, details of how many micrograms of protein were loaded per gel lane need to be added.

Reviewer #3 (Remarks to the Author):

The manuscript submission by Jianjie Ma and colleagues reports on the role of MG53/TRIM72 in regulating innate antiviral responses by controlling inflammatory responses to virus infection. Results from cell culture experiments and infection studies with MG53-knockout mice revealed that inflammatory responses are elevated in the absence of MG53 and following virus infection; however, the absence or presence of MG53 did not alter viral titers. Additional in vitro studies indicated that virus infection decreased MG53 protein levels by affecting translation, but not transcription. In general, the study presents findings for a novel role and potential mechanism by which MG53 regulates innate antiviral responses; however, the manuscript requires modifications to improve clarity before it could be considered for acceptance for publication.

Item 1. Considering the current findings, it would be relevant to describe whether MG53 expression and/or function is altered in patients diagnosed with type I interferonopathies or other inflammatory conditions?

Item 2. The scientific premise for the study is not adequately developed. The introduction summarizes that MG53 is generally a muscle-specific factor that functions in cell membrane repair. The rationale for suspecting MG53 would be involved in regulating inflammatory responses must be improved. The introduction would also benefit from inclusion of relevant references that strengthens the connection between innate immunity and tissue repair, especially by what was perceived to be a muscle-specific factor. As the in vitro studies included THP-1 cells, the connection of wound repair and inflammatory responses in macrophages, and thus the subject of investigation in Figure 1, is unclear.

Item 3. The authors demonstrated by in vitro studies that Sendai virus infection suppressed translation of MG53, which resulted in increased expression of interferon and IL-1beta. These in vitro studies were then followed by in vivo studies in which wild-type or MG53-deficient mice were infected by PR8 to examine alterations in morbidity and host responses. It is unclear if PR8 alters MG53 proteins levels in the same manner as by Sendai virus infection.

Item 4. Is the expression of other genes regulated in a calcium-dependent manner?

Item 5. For Figure 1C, was PMA included during the duration of the experiment? PMA treatment increased MG53 protein levels; however, it is unclear if MG53 protein levels decreased due to lack of stimulation by PMA or if virus infection actively decreased MG53 protein levels. For Figure 1D, were only two mock samples analyzed for the 48-hour time point? The figure legend should be revised to indicate what D1 and D2 designate.

Item 6. The title of the legend for Figure 6 indicated that MG53 regulated the activity of the ryanodine receptor. However, the data presented demonstrates an interaction between MG53 and the ryanodine receptor calcium channel protein. The figure does not provide data that MG53 specifically regulates the function of the ryanodine receptor calcium channel protein. The data presented in Figure 5 indicated that MG53 regulated intracellular calcium dynamics, and inclusion of the agonist 4-CmC indicated the involvement of the ryanodine receptor calcium channel protein. However, there is no data that indicated MG53 specifically targeted the function of the ryanodine receptor calcium channel.

Detailed Response to Reviewers' Critiques

Reviewer #1 (Remarks to the Author):

The authors are the leaders on the TRIM72 area of research. TRIM72 is also known as MG53. The new findings are truly exciting. TRIM72 is essential for cell membrane repair and is believed to be a muscle specific. The group now has convincing data to demonstrate that human macrophages express MG53, and that MG53 protein is reduced following viral infection. Knockdown of MG53 in macrophages leads to increased type I interferon (IFN) production upon infection, implicating MG53 as a negative regulator of IFN response. They further demonstrate a specific interaction into of MG53 signaling in activation of NFκB signaling, which is linked to an increase in intracellular calcium oscillation mediated by ryanodine receptor (RyR). Last, they show that MG53 interacts with RyR and inhibits IFNβ transcription in an RyR-dependent manner. This study establishes MG53 as a new macrophage target for control of virus-induced disease and tissue injury. It can have broad implications for public healthcare. I only have very minor comments:

a) Add a couple of sentences about the overall role of macrophages on inflammation and the switch in M1 to M2 for resolution of inflammation.

We thank the reviewer for his/her suggestion. We have taken this advice and added the following to the Introduction of the paper (page 3, line 40-45):

“Macrophages are innate immune cells of myeloid origin that exhibit a wide range of function and phenotypic heterogeneity. Macrophages are often classified as either canonically activated inflammatory macrophages (M1), or alternatively activated macrophages (M2) which are anti-inflammatory in nature and assist in tissue repair^{1,2}. As such, macrophages play a pivotal role in the response to and recovery from infectious pathogens such as viruses.”

b) Item a above, do the same in the discussion. Could speculate whether MG53 interaction occurs with M1 or M2, likely M2.

We have implemented the reviewer's suggestion by adding to the Discussion (page 10, line 296-304):

“*In vitro* studies, using macrophages infected with virus, mimic an inflammatory M1 activation of macrophages. Our findings show that suppression of MG53 exacerbates inflammatory cytokine secretion following viral infections, suggesting that MG53 functions to negatively regulate M1 macrophages. *In vivo* studies in which mice were infected with influenza virus supports the notion of MG53 suppressing M1 macrophages because they yield elevated inflammatory cytokine production in the absence of MG53. However, due to the exaggerated recovery time of MG53 knockout mice, it is possible that MG53 might also act to regulate M2 macrophage function and tissue repair. Further studies are required to fully demonstrate the impact of MG53 expression on different macrophage phenotypes and their unique functions.”

c) The data is very robust and it is excellent to see all data points plotted for transparency. We do recommend that for some of the data, the authors further check the statistics with non-parametric tests and ad hoc adjustments.

We thank the reviewer for bringing this to our attention. We have gone back and further evaluated data sets with non-parametric statistical testing to confirm findings in all figures when applicable.

d) The authors could test using either Pearson or Spearman whether there is a direct correlation between MG53 levels and immune activation or disease state.

The reviewer makes an interesting suggestion. Unfortunately, our data with WT and KO animals (i.e., MG53 present or absent) does not ideally lend itself to calculating correlations based on MG53 levels.

We have analyzed one study in which we have discovered a correlation between respiratory syncytial virus (RSV) severity in infants and MG53 expression (lower MG53 expression correlates with more severe RSV infection). The data derives from geo database series GSE77087 and the following

publication <https://www.ncbi.nlm.nih.gov/pubmed/27135599> which analyzed whole blood mRNA. While we could not find similar publicly available data with influenza virus infection, and while we do not intend to include this RSV analysis in our manuscript, we include this here for the reviewers as evidence that further analysis of the role of MG53 as a biomarker in humans is an interesting area of future investigation.

Reviewer #2 (Remarks to the Author):

In their paper the Authors describe a novel role for the skeletal muscle protein MG53 as a mediator of innate immune response. The Authors show that MG53 is expressed in human macrophages where it regulates the expression of interferon. Knocking down MG53 leads to increased interferon production via NFkB signaling. The increased nuclear localization of NFkB is mediated by RyR-dependent calcium oscillations. These observations are novel and interesting, especially because until now the primary role of MG53 was thought to be primarily tissue (skeletal muscle) repair. In my opinion additional experiments need to be performed in order to be convincing.

Major comments

1) In Figure 1 the Authors show by IP that macrophages express MG53. Can they see an immunopositive band by simple Western blot and not only after immunoprecipitation? No information regarding the total amount of protein loaded is indicated. How many cells were lysed? How many micrograms of proteins were loaded on the gels? They state that the levels are much lower than in skeletal muscle, but this is not quantified. Quantification by RT-qPCR or western blot comparing the intensity of the immunopositive band in skeletal muscle, cardiac muscle, macrophages and several other tissues is necessary. No explanation for panel 1D is included. How were the values normalized?

We appreciate the reviewer's detailed analysis. MG53 can be positively identified in macrophages by western blotting alone. Details regarding protein loading for all western blots have been added to the appropriate figure legends.

As suggested, we have compared MG53 protein expression between skeletal muscle, heart tissue, and macrophages via western blotting. The new data is presented as **Supplemental Fig. S2**.

We have revised the Results section to include citation of **Fig. 1D** (see page 5, line 109-110). Thank you.

2) In figure 2 and figure 3 they show that knocking down/out MG53 leads to an increase in IFN production, yet mice infected with influenza virus do not show a difference in viral load. How come? IFN is released by cells to counterbalance viral infections and thus higher levels of IFN should be associated with a decreased viral load and lower morbidity. Mice should recover faster and not slower if they have an enhanced inflammatory response. Production of additional cytokines should be assessed including IL6, TNFa, IL12, IL23 as well as chemokines. Phenotypically distinct classes of macrophages exist, M1 which are pro-inflammatory and microbicidal and M2 which are immunosuppressive. Does the absence of MG53 affect all

macrophage types equally? Please also add serum levels of CRP as this is a good indication of the inflammatory response.

The reviewer notes that IFN suppresses virus infection. While this is true, there are dozens of negative regulatory proteins that inhibit IFN induction and signaling pathways in order to dampen the response and minimize tissue-damaging effects of IFN. This dampening of the IFN response is beneficial because detrimental inflammatory effects of IFNs continue to accumulate beyond the point at which their antiviral effects peak. This is perhaps best demonstrated by the exacerbation of viral disease seen in animals that are knocked out for negative regulators of IFNs. This enhanced disease phenomenon caused by loss of IFN regulation is reviewed in detail by Porritt and Hertzog, *Trends in Immunology*, 2015.

Mice with deficiencies in type I IFN responses (e.g., STAT1, IFNAR, IFN KOs) show a 0.5-2 log increase in influenza virus lung titers (Garcia-Sastre, *J Virol*, 1998; Arimori, *Virology*, 2015; Koerner, *J Virol*, 2007). Given these nominal changes in viral titers in the most extreme IFN deficiencies, it is perhaps not surprising that *increasing* IFN production in MG53 KO mice beyond what is made in WT mice did not further suppress virus replication, but did correlate with detrimental effects consistent with enhanced inflammation (increased infiltration of lymphocytes into the tissue and longer recovery times post infection). Thus, increased inflammation is generally not beneficial in influenza virus recovery. Rather, the magnitude of the cytokine storm produced during infection is associated with levels of morbidity in both humans and mice (for relevant review articles on the topic of the influenza cytokine storm, see Guo, *Semin Immunopathol*, 2017, and Teijaro, *Curr Top Microbiol Immunol*, 2015). Our results are fully consistent with increased pathology due to enhanced inflammation during influenza virus infection.

To address this topic, we have revised the discussion to include the topic regarding the detrimental effects of IFNs, the numerous negative regulatory mechanisms for IFN pathways, and the effects of the cytokine storm in influenza virus infection. Please see page 10, line 270-274.

Our manuscript focuses on the role of MG53 in regulation of type I IFN production as well as other inflammatory cytokines. In addition to IFN and IL1b that were shown in the manuscript, we performed additional experiments in which we measured the inflammatory cytokine IL-6, which was increased in the lung and heart after infection. These data are now included as **Fig 3G** and fully support our previous conclusion that MG53 is a negative regulator of type I IFN and inflammatory responses during virus infection. Please see page 6, line 150-152.

Infections of macrophages with viruses will generally activate the cells toward an M1 phenotype with production of IFN and other inflammatory cytokines. In this inflammatory context, we have observed that MG53 is a negative regulator of cytokine production *in vitro*. We also observed a similar negative regulation of inflammation by MG53 in mouse infections. We agree that studies of anti-inflammatory/reparative M2 macrophages would also be interesting and that M2 macrophages likely play a role in lung recovery in our *in vivo* infection system. However, specific effects of MG53 on M2 polarization will be a future study beyond our current investigation of inflammatory responses in virus infection. These statements have been added to the revised discussion (page 10, line 296-304).

Given that influenza virus only infects specific organs (e.g., lung and heart) and does not cause systemic viremia, we did not collect serum samples in our studies. Further, while serum CRP is generally increased in severe bacterial infections, including sepsis, it is not a commonly used clinical biomarker for severity of influenza virus infections. Thus, we have relied on more direct measurements of cytokines in specific tissues in our studies.

3) The fact that lungs and heart show increased levels of IFN and IL1 may simply be a consequence of the higher number of macrophages present in those tissues. Can the levels of these cytokines be normalized to number of macrophages present? As shown in figure 3 levels of the cytokines are not normalized to anything.

We understand the reviewer's concern. Cytokine levels presented in **Fig. 3E-G** of our study are the actual concentrations within equal volumes of tissue homogenates. These measurements provide direct assessment of the physiological role of MG53 in modulation of inflammatory response following viral infection. The ~100% increase in IFN that we observe in infected MG50 KO lungs (**Fig. 3E**) is unlikely

explained by the ~15% increase in infiltration of CD45 positive cells (**Fig. 3C&D**).

Thus, the increased levels of IFN and IL1 β observed with the MG53 KO mice is unlikely a consequence of the higher number of immune cells present in those tissues. We have added the following sentence to the revised manuscript (page 6, line 154-159):

“Cytokine levels presented in **Fig. 3D-3G** are the actual concentrations within equal volumes of tissue homogenates. These measurements provide direct assessment of the physiological role of MG53 in modulation of inflammatory response following viral infection. The ~100% increase in IFN β that we observe in infected MG53 KO lungs (**Fig. 3E**) is unlikely explained by the ~15% increase in infiltration of CD45 positive cells (**Fig. 3D**). Thus, the increased levels of IFN β and inflammatory cytokines observed with the MG53 KO mice is unlikely a consequence of the higher number of immune cells present in those tissues”.

4) The involvement of RyR in calcium homeostasis in THP cells is not convincing. Extremely high levels (2.5 mM) of 4-CmC were used. The Authors need to show that there is an effect on calcium release following the addition of 300-600 μ M 4-CmC or 5-10 mM caffeine. They should also show that this effect can be blocked with dantrolene, procaine or by pre-incubating the cells with high levels of ryanodine as well as in cells silenced for RyR1 and RyR2.

We appreciate the reviewer's comments and suggestions. In the revised manuscript, we provide new data to show that 4-CmC induced Ca release from THP-1 cells can be completely blocked by dantrolene (see **Supplemental Fig. S3**). This finding further supports the role of RyR-mediated Ca release as a contributing factor for MG53's anti-inflammatory function associated with viral infection. Please see page 7, line 212-215 for description of the new result.

Through literature search, we found “earlier studies by Treves and colleagues demonstrated that mutations of RyR1 in human patients with malignant hyperthermia and central core disease impacted IL-6 release from skeletal muscle⁷⁶, which is consistent with the role of RyR-mediated calcium release in modulation of inflammatory cytokine release”. We have added this sentence to our revised Discussion. Please see page 11, line 326-328:

5) The qPCR results show that RYR1 and RYR2 are expressed in THP cells and that their levels of expression are not affected by MG53 levels. What are the levels of RYR1 and RYR2 expression in THP cells compared to heart, skeletal muscle, brain, cerebellum, smooth muscle cells and other immune cells?

The reviewer poses an excellent question. Immune cells, such as THP-1 cells, express much lower levels of RyR relative to striated muscle. The Human Protein Atlas consensus dataset reports that monocytes and dendritic cells express 2.7% and 2.3% RyR1 mRNA, respectively, compared to skeletal muscle (<https://www.proteinatlas.org/ENSG00000196218-RYR1/tissue>). While others have also reported low levels of RyR2 mRNA in THP1 cells (reference 10), precise expression level relative to heart tissue is unknown and requires further study. Literature suggests that RyR2 expression in immune cells is regulated by cytokine stimulation, such as TGF- β , and immune activation (reference 9). We have added the following sentences to the revised Discussion (page 11, line 320-325):

“The Human Protein Atlas consensus dataset reports that, relative to skeletal muscle, immune cells such as monocytes and dendritic cells express 2.7% and 2.3% of the amount of RyR1 RNA, respectively⁷⁵. Since knocking down the expression of MG53 impacted intracellular calcium oscillation in THP1 cells, in principle other calcium-dependent genes may also be altered which can contribute to the increased inflammatory response of the THP1 cells following viral infection. Additional genomic, biochemical, and functional studies will be required to address this topic”.

6) It is puzzling that MG53 and RyR1 form a complex since to my understanding MG53 is in a different compartment (either cytosolic or sarcolemmal) while the RyR1 is in the sarcoplasmic reticulum. Does MG53 form a complex with the DHPR? and junctophilin? Because MG53 is a TRIM protein, it may be sticky and bind non-specifically to many proteins in vitro but this may never happen in vivo.

The reviewer is correct that during membrane damage MG53 localizes to the sarcolemma of muscle fibers. However, under basal conditions MG53 associates with intracellular vesicles and multiple intracellular organelles including the sarcoplasmic reticulum and mitochondria. *Ahn et al* (reference 37) reported that MG53 can interact with the plasma membrane Ca channel, Orai1, but does not interact with DHPR in rabbit triad vesicle preparation. Further experiments are needed to determine whether MG53 associates with junctophilins. We agree with the reviewer's concern that even though our co-IP studies demonstrate the co-presence of MG53 and RyR1 in the triad complex of skeletal muscle, whether or not such MG53-RyR1 interaction is direct or require the participation of an intermediate protein component requires further studies. We have added this to our revised Results section. See page 8, line 242-244.

Minor

1) Statistical analysis tests are not always defined (see for example figure 4, 5, 7, which post hoc test was used after the ANOVA? What was used in figure 5F?).

We have gone back and appropriately defined all statistical tests in each figure legend.

2) Many panels in the figures are not described. Please add missing information to panels 1D, 2C.

We thank the reviewer for bringing this to our attention. We have added appropriate descriptions to figure panels in each figure legend.

3) For all experiments, details of how many micrograms of protein were loaded per gel lane need to be added.

We have added protein loading details to all western blot figures within their respective figure legends. Thank you for your suggestions.

Reviewer #3 (Remarks to the Author):

The manuscript submission by Jianjie Ma and colleagues reports on the role of MG53/TRIM72 in regulating innate antiviral responses by controlling inflammatory responses to virus infection. Results from cell culture experiments and infection studies with MG53-knockout mice revealed that inflammatory responses are elevated in the absence of MG53 and following virus infection; however, the absence or presence of MG53 did not alter viral titers. Additional in vitro studies indicated that virus infection decreased MG53 protein levels by affecting translation, but not transcription. In general, the study presents findings for a novel role and potential mechanism by which MG53 regulates innate antiviral responses; however, the manuscript requires modifications to improve clarity before it could be considered for acceptance for publication.

Item 1. Considering the current findings, it would be relevant to describe whether MG53 expression and/or function is altered in patients diagnosed with type I interferonopathies or other inflammatory conditions?

This is an excellent suggestion from the reviewer. We believe this is beyond the scope of the current study however and look forward to researching this in the future. **We thank the editor's support for addressing this topic as future studies.**

Item 2. The scientific premise for the study is not adequately developed. The introduction summarizes that MG53 is generally a muscle-specific factor that functions in cell membrane repair. The rationale for suspecting MG53 would be involved in regulating inflammatory responses must be improved. The introduction would also benefit from inclusion of relevant references that strengthens the connection between innate immunity and tissue repair, especially by what was perceived to be a muscle-specific factor. As the in vitro studies included THP-1 cells, the connection of wound repair and inflammatory responses in macrophages, and thus the subject of investigation in Figure 1, is unclear.

We appreciate the reviewer's comment and have worked to further clarify the rationale of the study by adding the following statements in the revised Introduction (page 3, line 74-81):

“The wide therapeutic effects of MG53 in multiple tissues and in conditions in which pathology is mediated by inflammatory immune responses suggest that MG53 may have additional functions beyond membrane repair. While MG53 protein is mainly present in skeletal and cardiac muscles, our previous studies have shown that the renal and alveolar epithelial cells also contain detectable amounts of MG53 that contribute to kidney and lung functions under physiological and pathological conditions^{24,25}. Here we present data showing that human monocytic THP1 cells, as well as primary human blood monocyte-derived macrophages express MG53 protein. Given the intimate association between innate immunity and tissue repair, we hypothesized that MG53 regulates inflammation during infection and tissue injury”.

Item 3. The authors demonstrated by in vitro studies that Sendai virus infection suppressed translation of MG53, which resulted in increased expression of interferon and IL-1beta. These in vitro studies were then followed by in vivo studies in which wild-type or MG53-deficient mice were infected by PR8 to examine alterations in morbidity and host responses. It is unclear if PR8 alters MG53 proteins levels in the same manner as by Sendai virus infection.

Per recommendation of the reviewer, we have conducted further studies to compare the effects of Sendai virus and PR8 infection in THP-1 cells. As shown in the new **Supplemental Fig. S2**, while Sendai virus infection consistently led to reduced MG53 protein levels in THP-1 cells, we did not observe a significant decrease in MG53 levels following PR8 infection. This indicates a virus-specific difference in the antagonism of MG53 levels. This observation, while interesting, is a peripheral finding that does not change the main conclusion of our work showing that MG53 negatively regulates cytokine production induced by virus infection.

The following text has been added to the revised manuscript (page 5, line 110-114):

“We also compared the effects of SeV and influenza virus H1N1 strain PR8 infection on MG53 expression in THP1 cells. As shown in **Supplemental Fig. S2**, while SeV infection consistently led to reduced MG53 protein levels in THP1 cells, influenza virus infection did not appear to induce a significant decrease in MG53 in THP1 cells. This indicates a potential virus-specific difference in the antagonism of MG53 expression in THP1 cells”.

Item 4. Is the expression of other genes regulated in a calcium-dependent manner?

To address the reviewer’s question, we have added the following paragraph to the revised Discussion (page 11, line 320-325):

It is known that calcium regulates the transcriptional activity of numerous genes via a process known as excitation-transcription coupling. Since knocking down the expression of MG53 impacted intracellular calcium oscillation in THP1 cells, in principle other calcium-dependent genes may also be altered which can contribute to the increased inflammatory response of the THP1 cells following viral infection. Further genomic, biochemical, and functional studies are required to address this topic.

Item 5. For Figure 1C, was PMA included during the duration of the experiment? PMA treatment increased MG53 protein levels; however, it is unclear if MG53 protein levels decreased due to lack of stimulation by PMA or if virus infection actively decreased MG53 protein levels. For Figure 1D, were only two mock samples analyzed for the 48-hour time point? The figure legend should be revised to indicate what D1 and D2 designate.

Sorry for not stating clearly the details in our original manuscript. PMA was included throughout the experiment, so the decrease in MG53 protein is not due to PMA. We have amended the figure legend to make this clear. In our experimental setups we included one control sample for comparison with 3 infected samples. The experiment was repeated three times at the 24 h time point and twice at the 48-h time point. Results from all experiments and at both time points are consistent.

Item 6. The title of the legend for Figure 6 indicated that MG53 regulated the activity of the ryanodine receptor. However, the data presented demonstrates an interaction between MG53 and the ryanodine receptor calcium channel protein. The figure does not provide data that MG53 specifically regulates the function of the ryanodine receptor calcium channel protein. The data presented in Figure 5 indicated that MG53 regulated intracellular calcium dynamics, and inclusion

of the agonist 4-CmC indicated the involvement of the ryanodine receptor calcium channel protein. However, there is no data that indicated MG53 specifically targeted the function of the ryanodine receptor calcium channel.

We agree with the reviewer that the title of **Figure 6** is misleading. Clearly, the main content of **Figure 6** is not related to MG53 regulation of ryanodine receptor activity; rather it mainly shows the presence of RyR mRNA in THP1 cells and its association with MG53 in the triad complex in skeletal muscle. Thus, we have amended the Figure legend as follows: "MG53 associates with ryanodine receptor".

REVIEWERS' COMMENTS:

Reviewer #2 (Remarks to the Author):

The Authors have answered by queries. The manuscript is now acceptable for publication.

Reviewer #3 (Remarks to the Author):

The reviewer appreciates the revisions to the manuscript. The authors have satisfactorily addressed the concerns of this reviewer.